# Reinforcement-aware Knowledge Distillation for LLM Reasoning

**Zhaoyang Zhang** [1]   **Shuli Jiang** [1]   **Yantao Shen** [1]   **Yuting Zhang** [1]   **Dhananjay Ram** [1]   **Shuo Yang** [1]   **Zhuowen Tu** [1]
**Wei Xia** [1]   **Stefano Soatto** [1]

## Abstract

Reinforcement learning (RL) post-training has recently driven major gains in long chain-of-thought reasoning large language models (LLMs), but the high inference cost of such models motivates distillation into smaller students. Most existing knowledge distillation (KD) methods are designed for supervised fine-tuning (SFT), relying on fixed teacher traces or teacher-student Kullback–Leibler (KL) divergence-based regularization. When combined with RL, these approaches often suffer from distribution mismatch and objective interference: teacher supervision may not align with the student's evolving rollout distribution, and the KL regularizer can compete with reward maximization and require careful loss balancing. To address these issues, we propose *RL-aware distillation* (RLAD), which performs *selective imitation* during RL—guiding the student toward the teacher only when it improves the current policy update. Our core component, *Trust Region Ratio Distillation* (TRRD), replaces the teacher-student KL regularizer with a PPO/GRPO-style likelihood-ratio objective anchored to a teacher–old-policy mixture, yielding advantage-aware, trust-region-bounded distillation on student rollouts and naturally balancing exploration, exploitation, and imitation. Across diverse logic reasoning and math benchmarks, RLAD consistently outperforms offline distillation, standard GRPO, and KL-based on-policy teacher-student knowledge distillation.

## 1. Introduction

Large Language Models (LLMs) have advanced rapidly in generation and reasoning capabilities through scaling, but the resulting inference cost makes deployment challenging under tight latency or budget constraints. This motivates model compression, with knowledge distillation (KD) (Hinton et al., 2015) as a standard approach: a smaller *student* model learns to imitate a larger *teacher* model, improving quality at fixed inference budgets, e.g., Llama 3.2 (Meta, 2024), DeepSeek-R1 (Guo et al., 2025). In the supervised fine-tuning (SFT) regime, KD for LLMs is commonly realized as: (i) **offline distillation** (Guo et al., 2025; Hui et al., 2024; Yang et al., 2025), where students are trained on teacher-generated trajectories; (ii) **off-policy distillation** (Ko et al., 2024; Gu et al., 2024), which matches teacher and student distributions (e.g., via KL divergence) on a fixed dataset; and (iii) **on-policy distillation** (Xu et al., 2025; 2024; Lin et al., 2020), which computes teacher–student divergence on student-generated rollouts. However, extending these SFT-oriented recipes to RL post-training is non-trivial: RL continually updates the policy from reward feedback, causing the rollout distribution to shift over time and making static or mismatched distillation targets suboptimal and potentially unstable within the RL loop.

Consequently, a common practice in distilling reasoning models trained with RL is to adopt a simplified pipeline: first train a strong teacher with RL, then perform *offline* distillation by SFT on the teacher-generated traces (e.g., DeepSeek-R1 distillation) (Guo et al., 2025). Although straightforward, this approach does not perform RL optimization for the student and provides no teacher guidance that adapts to the student's evolving policy, limiting its ability to transfer *adaptive* reasoning behaviors. To bridge this gap, recent work such as KDRL (Xu et al., 2025) integrates KD into the RL loop by adding an on-policy KL regularizer between teacher and student on student rollouts. This enables joint optimization, but introduces two practical challenges. First, the RL objective and the KL regularizer can compete, making performance sensitive to the relative weighting and requiring careful hyperparameter tuning. Second, teacher–student divergence is computed on student-generated trajectories that may lie outside the teacher's high-probability region, which can weaken the supervision signal and reduce fidelity to the teacher's reasoning behavior.

To address the above challenges, in this work, we propose *RL-aware distillation* (RLAD), a distillation frame-

[1]AWS Agentic AI. Correspondence to: Zhaoyang Zhang <ozhaozha@amazon.com>.

*Proceedings of the 43rd International Conference on Machine Learning*, Seoul, South Korea. PMLR 306, 2026. Copyright 2026 by the author(s).

work tailored to RL post-training of LLMs. The key principle of RLAD is *selective imitation*: the student is encouraged to follow the teacher only when doing so is beneficial for improving the current policy under the RL objective. Motivated by techniques of clipped trust-region policy optimization (Schulman et al., 2015; 2017; Shao et al., 2024), we introduce Trust Region Ratio Distillation (TRRD), which replaces standalone KL regularization with a PPO/GRPO-style likelihood-ratio objective anchored to a teacher–student old-policy mixture. With PPO/GRPO-style clipping, this yields advantage-aware, trust-region-bounded distillation on student rollouts. As a result, RLAD allows the student to move toward high-advantage behavior within the anchor distribution while opportunistically exploiting teacher guidance, implicitly balancing exploration, exploitation, and imitation during RL training.

We evaluate RLAD on both math and logic reasoning benchmarks across varying task scales and model sizes (0.5B–7B students; up to a 32B teacher), and observe consistent improvements over prior RL-based distillation baselines across both domains. On logical reasoning with the K&K Logistics dataset (Xie et al., 2025a) using a Qwen3-8B teacher, RLAD substantially improves average accuracy: for Qwen3-0.6B, from 0.76 to 0.94 at 8K context (and $0.70{\rightarrow}0.90$ at 2K); for Qwen3-1.7B, from 0.95 to 0.99 at 8K (and $0.86{\rightarrow}0.93$ at 2K). RLAD also converges faster and attains higher rewards than both GRPO and KDRL, with the largest gains concentrated on the hardest subsets (PPL7 and PPL8), where it maintains a clear margin over KDRL. On long-context math reasoning with Qwen3-8B/Qwen3-32B teachers, RLAD improves the average performance (over five test sets and nine metrics) by +2.5 (Qwen3-1.7B-Base) and +5.5 (Qwen3-8B-Base) at 30K, and by +6.6/+2.6 for post-trained models under an 8K response budget compared to standard GRPO, consistently outperforming explicit-KL-style distillation and offline distillation. Gains are most pronounced on challenging benchmarks (e.g., AIME24/25 under Pass@32), while simpler benchmarks such as MATH500 show smaller improvements, suggesting performance saturation and model capacity limits.

Compared with KDRL on complex math reasoning, we observe two notable differences: (1) **RL behavior.** KDRL tends to produce larger relative gains in Pass@32 than in Pass@1 (Table 2), suggesting that its improvements are driven primarily by teacher imitation rather than reward-driven policy improvement; this is consistent with (Yue et al., 2025a), which notes that RL typically benefits Pass@1 more than large-$K$ metrics. (2) **Training stability.** KDRL also exhibits less stable validation dynamics, with pronounced oscillations in Figure 2. In contrast, RLAD yields more stable training and stronger Pass@1 gains, indicating a better balance between imitation and reinforcement learning.

**In summary,** our contributions are:

1. We propose *RL-Aware Distillation (RLAD)* (Section 4), a new framework for incorporating teacher guidance into RL post-training via *Trust Region Ratio Distillation (TRRD)* (Section 4.1), a clipped importance-ratio objective anchored on a teacher–old student mixture.
2. We provide in-depth analysis of TRRD (Section 4.1), demonstrating that it enables reward-conditioned, selective imitation while preserving trust-region–bounded and stable policy updates.
3. Through extensive experiments on logical and long-context mathematical reasoning (Section 5), we demonstrate that RLAD consistently outperforms GRPO, explicit KL-based distillation, and offline distillation, achieving larger gains on harder benchmarks and improved training stability.

## 2. Related Work

**Reinforcement Learning for Reasoning LLMs.** The recent success of Kimi (Team et al., 2025b;a) and DeepSeek-R1 (Guo et al., 2025) demonstrates that reinforcement learning (RL) can effectively post-train long-chain-of-thought (long-CoT) reasoning LLMs. Building on this progress, RL-based methods for reasoning can be broadly grouped into three lines: (i) applying RL directly to pretrained base models, exemplified by R1-Zero-style training (Xie et al., 2025b; Hu et al., 2025; Zeng et al., 2025); (ii) post-training smaller distilled models to push the reasoning frontier under limited capacity (Luo et al., 2025; He et al., 2025; Wen et al., 2025; Mei et al., 2025); and (iii) developing backbone-agnostic algorithmic or strategy improvements that are largely decoupled from a specific model family (Yu et al., 2025; Liu et al., 2025; Zhang & Zuo, 2025; Yue et al., 2025b).

**Knowledge Distillation for LLMs.** Knowledge distillation (KD) has become a core approach for transferring capabilities from large teacher LLMs to smaller students (Gu et al., 2024; Agarwal et al., 2024b; Ko et al., 2024). In contrast to BERT-style distillation that often relies on intermediate-layer matching (Wang et al., 2020; Ko et al., 2023), LLM distillation predominantly adopts output-level supervision—especially token-level logit matching—since intermediate representations are difficult to align across scales and architectures. We categorize LLM distillation methods by *how training data is collected* and *which policy induces the student's learning signal*.

*(1) Offline distillation.* Offline distillation constructs a fixed teacher-labeled dataset (often from instruction-tuned or RL-trained reasoning teachers) and trains the student via standard supervised fine-tuning (SFT) on teacher-generated traces. This paradigm is widely used in reasoning distillation due to its simplicity and stability (Guo et al., 2025; Ye

et al., 2025; Muennighoff et al., 2025; Tian et al., 2025). Notably, offline distillation is a *pure imitation* process: it does not require teacher queries during student training.

*(2) Off-policy KD.* Off-policy KD still leverages teacher guidance, but computes the supervision signal (e.g., teacher logits or preferences) on data not generated by the *current* student policy. This includes standard logit-based distillation on static prompts or teacher-sampled sequences, performed at the sequence level (Hsieh et al., 2023; Guo et al., 2025) or token-logit level (Liu et al., 2024; Gu et al., 2025). Recent work further refines this setting: (Li et al., 2025) mitigates long-tail noise using top-$k$ teacher/student logits and exploits ranking structure via logit differences, while (Zhang et al., 2024) introduces dual-space knowledge distillation (DSKD) to better align teacher and student output representations. Other extensions study efficiency and training improvements in related distillation settings (Xu et al., 2024; Zhang et al., 2024; Li et al., 2025).

*(3) On-policy KD.* On-policy KD trains the student on *student-generated outputs* (SGO) augmented with teacher feedback (e.g., teacher logits, divergences, or critiques). By learning from its own rollouts, on-policy KD can reduce exposure bias and better match the student's test-time distribution. Representative approaches include policy-gradient-based KD designed to reduce high-variance updates (Gu et al., 2024), and explicit SGO frameworks optimizing divergence objectives such as RKLD and JSD (Ko et al., 2024; Agarwal et al., 2024b). Recent technical reports also explore on-policy distillation for reasoning models and demonstrate its potential at scale (Yang et al., 2025).

**Distillation in RL Training.** Several works combine reward optimization with teacher supervision. Some modify preference objectives (e.g., DPO variants) to incorporate teacher signals (Liu et al., 2024; Pan et al., 2025). GKD (Agarwal et al., 2024a) provides a generalized on-policy distillation view; LUFFY (Yan et al., 2025) mixes off-policy trajectories with policy shaping for reasoning; and KDRL (Xu et al., 2025) jointly optimizes on-policy rewards with token-level KL distillation.

**Position of our RLAD.** Offline distillation is SFT and does not reflect RL post-training dynamics. Existing off-/on-policy KD in RL typically adds a explicit-KL-style auxiliary term, creating a multi-objective trade-off that can conflict with reward maximization. In contrast, RLAD folds teacher guidance into a trust-region for RL, so imitation is applied *only when* it benefits the current RL update, reducing brittle tuning between imitation and exploration.

## 3. Background and Notations

**Problem Statement.** Let $\mathcal{X}$ denote the space of prompts and $\mathcal{V}$ the vocabulary. We consider autoregressive langauge

models, which defines a stochastic policy $\pi$ parameterized by $\theta$, as $\pi_\theta(y|x) = \prod_{t=1}^{T} \pi_\theta(y_t|x, y_{<t})$, where $x \in \mathcal{X}$, $y = (y_1, y_2, \ldots, y_T) \in \mathcal{V}^T$ is the generated response. Define $y_{<t} = (y_1, \ldots, y_{t-1})$. Let $R : \mathcal{X} \times \mathcal{V}^* \to \mathbb{R}$ denote the reward function. Let the KL divergence be $\mathrm{KL}(\pi_{\theta_1} || \pi_{\theta_2}) = \mathbb{E}_{y_t \sim \pi_{\theta_1}} [\log \pi_{\theta_1}(y_t|x, y_{<t}) - \log \pi_{\theta_2}(y_t|x, y_{<t})]$.

In this work, we consider a *student–teacher* setting consisting of a smaller student model (policy) $\pi_{\theta^S}$ and a more powerful teacher model (policy) $\pi_{\theta^T}$. Our objective is twofold. First, following the standard reinforcement learning formulation, the student model is trained to maximize the expected reward: $\max_{\theta^S} J(\theta^S) = \mathbb{E}_{x \sim \mathcal{D}, y \sim \pi_{\theta^S}(\cdot|x)} [R(x, y)]$, where $\mathcal{D}$ denotes a distribution over prompts. Second, beyond reward maximization, we aim to transfer knowledge from the teacher model to the student, encouraging the student policy to align with the teacher's behavior while optimizing the task reward.

**Group Relative Policy Optimization (GRPO).** One popular RL algorithm for training LLMs to improve their reasoning capabilities is GRPO (Shao et al., 2024). Given a prompt $x \in \mathcal{X}$, GRPO first samples a group of $G$ responses from the current student policy $\pi_{\theta^S}$, denoted by $\{y^{(i)}\}_{i=1}^{G}$. Each response $y^{(i)}$ is assigned a scaler reward $r^{(i)}$. To normalize rewards at the prompt level, GRPO computes the mean and standard deviation of the group rewards on prompt $x$: $\mu(x) = \frac{1}{G} \sum_{i=1}^{G} r^{(i)}$ and $\sigma(x) = \sqrt{\frac{1}{G} \sum_{i=1}^{G} (r^{(i)} - \mu(x))^2}$. Let $\pi_{\theta^{S,old}}$ denote a fixed, earlier version of the student policy. For the $t$-th token of the $i$-th response, the importance sampling ratio is defined as

$$r_{i,t}^{\mathrm{GRPO}}(\theta^S) = \frac{\pi_{\theta^S}(y_t^{(i)} \mid x, y_{<t}^{(i)})}{\pi_{\theta^{S,old}}(y_t^{(i)} \mid x, y_{<t}^{(i)})}$$

and the GRPO objective is then

$$J^{\mathrm{GRPO}}(\theta^S) = \mathbb{E}_{x \sim \mathcal{D}, \{y_i\}_{i=1}^{G} \sim \pi^{S,old}(\cdot|x)} \tag{1}$$

$$\left[ \frac{1}{G} \sum_{i=1}^{G} \frac{1}{|y^{(i)}|} \sum_{t=1}^{|y^{(i)}|} \left\{ \min \left( r_{i,t}^{\mathrm{GRPO}}(\theta^S) \widehat{A}_{i,t}, \right.\right.$$

$$\left.\left. \mathrm{Clip}(r_{i,t}^{\mathrm{GRPO}}(\theta^S), 1 - \epsilon, 1 + \epsilon) \widehat{A}_{i,t} \right) \right\} \right] - \beta \mathrm{KL}(\pi_{\theta^S} || \pi_{\theta^{ref}})$$

where the group-relative advantage is $\widehat{A}_{i,t} = \frac{r^{(i)} - \mu(x)}{\sigma(x)}$. Here, $\epsilon$ denotes the GRPO-style clipping threshold, $\beta$ controls the strength of the regularization, and $\pi_{\theta^{ref}}$ is a reference policy kept fixed during training. In practice, $\pi_{\theta^{ref}}$ is typically obtained from a supervised fine-tuned model prior to the post-training RL stage.

**KDRL (Xu et al., 2025).** In the knowledge distillation (KD) setting, a straightforward approach to incorporating a teacher model is to directly replace the reference policy $\pi_{\theta^{ref}}$ in KL regularization term in the GRPO objective (1) with

the teacher policy $\pi_{\theta^T}$. The new KL regularization term becomes

$$
\begin{aligned}
&\mathrm{KL}(\pi_{\theta^S} || \pi_{\theta^T}) \hspace{3cm} (2)\\
&= \mathbb{E}_{y_t \sim \pi_{\theta^S}} \left[ \log \pi_{\theta^S}(y_t | x, y_{<t}) - \log \pi_{\theta^T}(y_t | x, y_{<t}) \right]
\end{aligned}
$$

Under this formulation, the student policy $\pi_{\theta^S}$ is explicitly regularized towards the teacher, and we denote the resulting objective by $J^{\mathrm{KDRL}}(\theta^S)$. Notably, the KL regularization term (2) coincides with the reverse KL distillation objective commonly used in classical KD methods (Xu et al., 2025), including supervised fine-tuning (SFT)–based distillation for large language models.

## 4. RL-Aware Distillation (RLAD)

Naive teacher-based distillation methods such as KDRL (Xu et al., 2025) incorporate the teacher policy $\pi_{\theta^T}$ via a fixed KL regularization term that is independent of the estimated advantage $\widehat{A}$. As a result, teacher guidance uniformly influences the student policy, including in regions where the teacher's behavior is misaligned with high-reward trajectories. This can lead to imitation that degrades task performance. Ideally, teacher guidance should be *selective*, reinforcing student updates only when the teacher's preferences are consistent with the reward signal. RLAD is designed to achieve this by integrating teacher information directly into the advantage-weighted policy update, rather than treating it as a static prior.

### 4.1. Trust Region Ratio Distillation (TRRD)

To address teacher–advantage misalignment, we propose Trust Region Ratio Distillation (TRRD), which incorporates the teacher policy into the GRPO importance ratio instead of imposing it through an unconditional KL penalty.

Recall that in GRPO, policy updates are governed by the importance ratio between the current student policy $\pi_{\theta^S}$ and a fixed previous policy $\pi_{\theta^{S,old}}$, ensuring alignment with the estimated advantage. TRRD generalizes this mechanism by interpolating between on-policy improvement and teacher guidance at the token level, controlled by a tunable coefficient $\alpha \in [0, 1]$.

Formally, for the $t$-th token of the $i$-th sampled response, we define the TRRD ratio as

$$
\begin{aligned}
r_{i,t}^{\mathrm{TRRD}}(\theta^S) &= \left( r_{i,t}^{\mathrm{GRPO}}(\theta^S) \right)^\alpha \left( r_{i,t}^{\mathrm{T}}(\theta^S) \right)^{1-\alpha} \quad (3)\\
&:= \left( \frac{\pi_{\theta^S}(y_t^{(i)}|x, y_{<t}^{(i)})}{\pi_{\theta^{S,old}}(y_t^{(i)}|x, y_{<t}^{(i)})} \right)^\alpha \left( \frac{\pi_{\theta^S}(y_t^{(i)}|x, y_{<t}^{(i)})}{\pi_{\theta^T}(y_t^{(i)}|x, y_{<t}^{(i)})} \right)^{1-\alpha}
\end{aligned}
$$

Replacing the GRPO importance ratio with $r_{i,t}^{\mathrm{TRRD}}$ yields the RLAD objective:

$$
J^{\mathrm{RLAD}}(\theta^S) = \mathbb{E}_{x \sim \mathcal{D}, \{y_i\}_{i=1}^G \sim \pi^{S,old}(\cdot|x)} \quad (4)
$$

$$
\left[ \frac{1}{G} \sum_{i=1}^G \frac{1}{|y^{(i)}|} \sum_{t=1}^{|y^{(i)}|} \left\{ \min \left( r_{i,t}^{\mathrm{TRRD}}(\theta^S) \widehat{A}_{i,t}, \right.\right.
$$

$$
\left.\left. \mathrm{Clip}(r_{i,t}^{\mathrm{TRRD}}(\theta^S), 1-\epsilon, 1+\epsilon) \widehat{A}_{i,t} \right) \right\} \right] - \beta \mathrm{KL}(\pi_{\theta^S} || \pi_{\theta^{\mathrm{ref}}})
$$

**TRRD as a Trust-Region Update.** Define the token-level mixture anchor ratio

$$
r_{\pi^{\mathrm{mix}}}(y_t | x, y_{<t}) := (\pi_{\theta^{S,old}})^\alpha (\pi_{\theta^T})^{1-\alpha}
$$

Then the TRRD ratio can be written as

$$
r_{i,t}^{\mathrm{TRRD}}(\theta^S) = \frac{\pi_{\theta^S}(y_t^{(i)} | x, y_{<T}^{(i)})}{r_{\pi^{\mathrm{mix}}}(y_t^{(i)} | x, y_{<T}^{(i)})}
$$

Thus, RLAD performs a PPO-style update with the trust region centered on $r_{\pi^{\mathrm{mix}}}$ rather than the previous student policy alone. Applying clipping bounds $r_{i,t}^{\mathrm{TRRD}} \in [1-\epsilon, 1+\epsilon]$ directly bounds the change in log-probability:

$$
|\log \pi_{\theta^S}(y_t^{(i)} | x, y_{<t}^{(i)}) - \log r_{\pi^{\mathrm{mix}}}(y_t^{(i)} | x, y_{<t}^{(i)})| \leq \log(1+\epsilon)
$$

which in turn constrains the KL divergence between the updated student policy and the mixture anchor at the token level. When $\alpha = 0$, TRRD recovers standard GRPO; when $\alpha = 1$, the trust region is fully teacher-anchored, yielding an objective closely related to DPO. Appendix C analyzes this interpolation in detail.

**TRRD as an Implicit Weighted KL Regularization.** Taking logarithms of TRRD ratio (3) yields

$$
\begin{aligned}
\log r_{i,t}^{\mathrm{TRRD}} &= \alpha(\log \pi_{\theta^S} - \log \pi_{\theta^{S,old}}) \quad (5)\\
&+ (1-\alpha)(\log \pi_{\theta^S} - \log \pi_{\theta^T})
\end{aligned}
$$

showing that *TRRD implicitly optimizes a convex combination of two KL terms: a trust-region term relative to the previous student policy, and a distillation term relative to the teacher policy*. Equivalently, TRRD can be viewed as optimizing a reward-augmented objective of the form:

$$
\begin{aligned}
\max_{\theta^S} \mathbb{E}[R(x, y)] &- \alpha \mathrm{KL}(\pi_{\theta^S} || \pi_{\theta^{S,old}})\\
&- (1-\alpha)\mathrm{KL}(\pi_{\theta^S} || \pi_{\theta^T})
\end{aligned}
$$

up to GRPO-style clipping and advantage normalization. In general, TRRD does not impose teacher imitation as a fixed constraint. Instead, it treats teacher alignment as a soft regularizer that competes with on-policy improvement.

**Why TRRD Fixes Teacher-Advantage Misalignment.** TRRD resolves this misalignment by embedding the teacher

policy *inside the advantage-weighted importance ratio*, so that teacher influence is modulated by the same mechanism that governs policy improvement in GRPO.

Specifically, for the $t$-th token of the $i$-th sampled response, following the TRRD ratio in (3), the policy gradient induced by the unclipped term in $r_{i,t}^{\text{TRRD}}(\theta^S)\widehat{A}_{i,t}$ is proportional to

$$\widehat{A}_{i,t}\left[\nabla_{\theta^S}\log\pi_{\theta^S}(y_t^{(i)}\mid x, y_{<t}^{(i)}) - \nabla_{\theta^S}\log r_{\pi^{\text{mix}}}(y_t^{(i)}\mid x, y_{<t}^{(i)})\right]$$

. Since $r_{\pi^{\text{mix}}}$ does not dependent on $\theta^S$, teacher influence affects the update only through the scaling of the student's log-probability by $\widehat{A}_{i,t}$. This structure yields three regimes:

1. Positive advantage ($\widehat{A}_{i,t} > 0$). Under positive advantage, the gradient encourages increasing $\log\pi_{\theta^S}$ relative to $r_{\pi^{\text{mix}}}$, subject to the clipping constraint on the TRRD ratio $r_{i,t}^{\text{TRRD}} = \pi_{\theta^S}/r_{\pi^{\text{mix}}}$.

   If the teacher assigns high probability to $y_t^{(i)}$, the mixture anchor $r_{\pi^{\text{mix}}}$ is larger, causing $r_{i,t}^{\text{TRRD}}$ to grow more slowly as $\pi_{\theta^S}$ increases. Consequently, the clipping threshold $(1 + \epsilon)$ is reached later, allowing a larger increase in the student's probability before the update saturates.

   If the teacher assigns low probability to $y_i^{(t)}$, $r_{\pi^{\text{mix}}}$ is smaller and the ratio reaches the clipping bound earlier, restricting the magnitude of the update even though the advantage is positive.

2. Negative advantage ($\widehat{A}_{i,t} < 0$). When the estimated advantage is negative, the objective encourages decreasing the student probability $\pi_{\theta^S}$, again subject to the clipping constraint on $r_{i,t}^{\text{TRRD}} = \pi_{\theta^S}/r_{\pi^{\text{mix}}}$.

   If the teacher assigns high probability to $y_i^{(t)}$, the mixture anchor policy ratio $r_{\pi^{\text{mix}}}$, which shifts the lower clipping threshold $r_{i,t}^{\text{TRRD}} \geq 1 - \epsilon$ to correspond to a smaller absolute decrease in $\pi_{\theta^S}$. As a result, the student's probability for $y_t^{(i)}$ cannot be reduced aggressively, preventing the policy from moving sharply away from actions strongly preferred by the teacher.

   Conversely, if the teacher assigns low probability to $y_t^{(i)}$, $r_{\pi^{\text{mix}}}$ is smaller and the lower clipping bound permits a larger relative decrease in $\pi_{\theta^S}$. In this case, the student is allowed to suppress the action more strongly, consistent with both the negative advantage signal and the teacher's preference.

3. Near-zero advantage ($\widehat{A}_{i,t} \approx 0$). The contribution of the term vanishes regardless of the teacher policy, ensuring that teacher influence is negligible when the reward signal is uninformative.

**Why TRRD enables better distillation over explicit-KL distillation.** TRRD can be interpreted as optimizing a reward-regularized objective with a convex combination of KL divergences to the previous student policy and the teacher policy. Unlike naive distillation, TRRD adaptively adjusts the trust region based on teacher agreement while preserving the reward-determined update direction, activating teacher influence only when supported by the advantage signal and the students are away from teacher trust-region, thereby balancing exploration, exploitation, and imitation.

# 5. Experiments

**Tasks.** We validate the effectiveness of RLAD on two representative reasoning settings: logical reasoning and mathematical reasoning.

**Baselines.** We compare RLAD against the following baselines: (i) **GRPO** (Shao et al., 2024), which optimizes the student using the GRPO objective (1) without teacher guidance (see Section 3). (ii) **KDRL** (Xu et al., 2025), which augments GRPO with a KL distillation term from a larger teacher (see Section 3). (iii) **SFT**: for complex math reasoning, following (Xu et al., 2025), which follows (Xu et al., 2025) and applies supervised fine-tuning on reject-sampled teacher outputs as an offline distillation baseline for complex math reasoning.

**Stability Considerations and Hyperparameter Ablations.** In KDRL, the KL regularization term between the student and teacher policies can become excessively large early in training, when the two policies are substantially mismatched, which may lead to unstable optimization. In contrast, RLAD anchors updates to a mixture of the teacher and previous student policies and enforces a trust region by clipping the TRRD ratio, which mitigates such outliers by construction. We further conduct ablation studies on the mixing coefficient $\alpha$ (Table 4 and Appendix A), finding that RLAD is largely insensitive to $\alpha$ across both mathematical and logical reasoning tasks, except at extreme values where one component dominates. Based on these results, we fix $\alpha = 0.5$ in all subsequent experiments.

## 5.1. Logical Reasoning Task

### 5.1.1. EXPERIMENTAL SETUP

**Training.** Following Logic-RL (Xie et al., 2025b), we conduct RL post-training for logical reasoning on the K&K dataset (Xie et al., 2025a). We adopt Qwen3-0.6B and Qwen3-1.7B as student models and optimize them with GRPO (Shao et al., 2024) on 8 H200 GPUs. Unless stated otherwise, we use a group size of 8, a learning rate of $1 \times 10^{-6}$, and GRPO clipping thresholds of 0.20 (lower) and 0.28 (upper). Training uses a micro-batch size of 32 and a global batch size of 256, with a maximum generation length of either 2,048 or 8,192 tokens depending on the setting. For teacher-guided methods (KDRL and RLAD), we use

| Model | Context Length | Training | PPL2 | PPL3 | PPL4 | PPL5 | PPL6 | PPL7 | PPL8 | Avg |
|---|---|---|---|---|---|---|---|---|---|---|
| o3-mini-high | 8K | - | 0.99 | 0.98 | 0.97 | 0.95 | 0.94 | 0.89 | 0.83 | 0.94 |
| Deepseek-R1 | 8K | - | 0.91 | 0.73 | 0.77 | 0.78 | 0.75 | 0.88 | 0.83 | 0.81 |
| Qwen3-0.6B | 8K | - | 0.77 | 0.45 | 0.52 | 0.47 | 0.32 | 0.34 | 0.22 | 0.44 |
| Qwen3-0.6B | 8K | GRPO | 0.86 | 0.85 | 0.85 | 0.76 | 0.72 | 0.66 | 0.63 | 0.76 |
| Qwen3-0.6B | 8K | KDRL (Xu et al., 2025) | **1** | 0.97 | **1** | **0.99** | **0.96** | 0.82 | 0.75 | 0.92 |
| Qwen3-0.6B | 8K | RLAD (ours) | **1** | **0.99** | 0.99 | 0.96 | 0.93 | **0.91** | **0.83** | **0.94** |
| Qwen3-1.7B | 8K | - | 0.95 | 0.99 | 0.90 | 0.80 | 0.77 | 0.85 | 0.70 | 0.85 |
| Qwen3-1.7B | 8K | GRPO | 1 | 0.99 | 1 | 0.98 | 0.95 | 0.92 | 0.81 | 0.95 |
| Qwen3-1.7B | 8K | KDRL (Xu et al., 2025) | 0.99 | **1** | **1** | 0.96 | **0.97** | 0.93 | 0.88 | 0.96 |
| Qwen3-1.7B | 8K | RLAD (ours) | **1** | **1** | **1** | **1** | **0.97** | **0.95** | **0.98** | **0.99** |
| Qwen3-0.6B | 2K | - | 0.69 | 0.47 | 0.41 | 0.22 | 0.09 | 0.08 | 0.03 | 0.28 |
| Qwen3-0.6B | 2K | GRPO | 0.84 | 0.83 | 0.85 | 0.76 | 0.56 | 0.58 | 0.50 | 0.70 |
| Qwen3-0.6B | 2K | KDRL (Xu et al., 2025) | **1** | 0.97 | **1** | 0.95 | 0.82 | 0.72 | 0.54 | 0.86 |
| Qwen3-0.6B | 2K | RLAD(ours) | **1** | **1** | 0.99 | **0.96** | **0.86** | **0.81** | **0.69** | **0.90** |
| Qwen3-1.7B | 2K | - | 0.90 | 0.70 | 0.66 | 0.43 | 0.21 | 0.16 | 0.05 | 0.44 |
| Qwen3-1.7B | 2K | GRPO | **1** | 0.98 | 0.97 | 0.90 | 0.87 | 0.79 | 0.52 | 0.86 |
| Qwen3-1.7B | 2K | KDRL (Xu et al., 2025) | **1** | 0.98 | **1** | **0.98** | 0.86 | 0.84 | 0.69 | 0.91 |
| Qwen3-1.7B | 2K | RLAD(ours) | 0.98 | **0.99** | **1** | **0.98** | **0.93** | **0.86** | **0.75** | **0.93** |

*Table 1.* Accuracy on the K&K Logistics testing subsets at 2K and 8K context length.

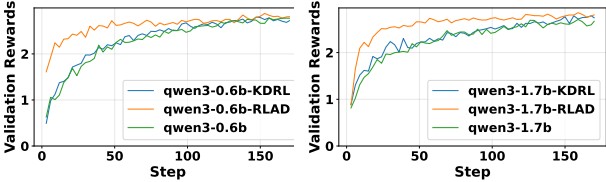

*Figure 1.* Comparisons of training dynamics RL training on logistics reasoning in terms of both training and validation average rewards for Qwen3-0.6B abd Qwen3-1.7B model.

Qwen3-8B as the teacher model.

**Dataset and Setup.** We evaluate all models on the test set of K&K Logistics dataset (Xie et al., 2025a), which is partitioned into difficulty-based subsets PPL2 - PPL8, where PPL2 and PPL8 are the easiest and hardest. We report accuracy on each subset. Unless otherwise specified, decoding uses a temperature of 0.6 and top-$p$ of 0.95, with a maximum generation length of 2,048 or 8,192 tokens, matching the corresponding training configuration.

### 5.1.2. MAIN RESULTS

**Training dynamics.** Figure 1 illustrates the training and validation reward trajectories during RL on the logical reasoning task for Qwen3-0.6B and Qwen3-1.7B. As shown, RLAD achieves faster convergence and higher final rewards than both vanilla GRPO training and teacher-based distillation with KDRL.

**Results.** Tables 1 shows that RLAD delivers consistent improvements across context lengths and student sizes. With

an 8K context window, RLAD improves the average accuracy of Qwen3-0.6B by +18% (0.94 vs. 0.76 under GRPO) and Qwen3-1.7B by +4% (0.99 vs. 0.95), outperforming KDRL in both settings With a 2K context window, the gains remain substantial: +20% for Qwen3-0.6B (0.90 vs. 0.70) and +7% for Qwen3-1.7B (0.93 vs. 0.86).

Notably, RLAD yields larger benefits on harder subsets. On the most challenging PPL8 split, RLAD improves Qwen3-0.6B from 0.63 to 0.83 at 8K and from 0.50 to 0.69 at 2K, while Qwen3-1.7B improves from 0.88 to 0.98 at 8K and from 0.52 to 0.75 at 2K (all compared to GRPO). Across these difficult regimes, RLAD also maintains a 6%–15% advantage over KDRL.

### 5.2. Complex Math Reasoning Task

#### 5.2.1. EXPERIMENTAL SETUP

**Training.** We follow the Skywork-OR1 training recipe (He et al., 2025) to evaluate RLAD on complex mathematical reasoning. We train RLAD and all baselines on the Skywork-OR1 training set (105K samples), enabling rejection sampling and adaptive entropy throughout RL training as in (He et al., 2025). Unless stated otherwise, we use a learning rate of $1 \times 10^{-6}$, a maximum prompt length of 2,048 tokens, and a maximum response length of 8K–30K tokens depending on the setting. Training uses a micro-batch size of 16 and a global batch size of 256. All runs are executed on 64 NVIDIA H200 GPUs for up to 5 epochs, and we select the checkpoint with the best validation performance on AIME24 (AI-MO, 2024a).

**Student model settings.** To assess RLAD under differ-

| Model | Training | Context | AIME24 | AIME24-Pass@32 | AMC23 | AMC23-Pass@32 | MATH500 | AMC24 | AMC24-Pass@32 | AIME25 | AIME25-Pass@32 | Avg |
|---|---|---|---|---|---|---|---|---|---|---|---|---|
| Qwen3-1.7B-Base | - | 30K | 2.1 | 20.8 | 16.3 | 70.8 | 29.8 | 8.2 | 44.6 | 0.9 | 6.2 | 22.2 |
| Qwen3-1.7B-Base | GRPO | 30K | 10 | 30.8 | 41.2 | **83.8** | 54.3 | 19.7 | 58.6 | 5.7 | 24.2 | 36.5 |
| Qwen3-1.7B-Base | KDRL | 30K | 10.2 | 32.8 | 41.1 | 83.6 | 56.2 | 21.0 | 59.7 | 6.2 | 24.4 | 37.2 |
| Qwen3-1.7B-Base | RLAD (ours) | 30K | **12.1** | **35.1** | **42** | 83.7 | **58.1** | **24.4** | **62.4** | **8.3** | **24.8** | **39.0** |
| Qwen3-8B-Base | - | 30K | 13.9 | 43.6 | 30.5 | 88.6 | 42.1 | 18.8 | 64.6 | 4.7 | 24.1 | 36.8 |
| Qwen3-8B-Base | SFT | 30K | 26.7 | 65.8 | 67.9 | 92.8 | 67.5 | 48.2 | 72.9 | 20.8 | 41.5 | 56.0 |
| Qwen3-8B-Base | GRPO | 30K | 37.7 | 77.8 | 72.9 | 94.8 | 68.2 | 49.2 | 76.9 | 22.8 | 48.5 | 61.0 |
| Qwen3-8B-Base | KDRL | 30K | 42.1 | **86.1** | 72.8 | 95.8 | 68.4 | 49.1 | 77.0 | 27.8 | 65.2 | 64.9 |
| Qwen3-8B-Base | RLAD (ours) | 30K | **47.8** | 85.4 | **73.6** | **98** | **68.6** | **49.8** | **77.4** | **31.7** | **66.4** | **66.5** |

*Table 2.* Long-context math reasoning results for Qwen3-Base models at 30K context length. Results are reported in terms of Pass@1 and Pass@32.

| Model | Training | Context | AIME24 | AIME24-Pass@32 | AMC23 | AMC23-Pass@32 | MATH500 | AMC24 | AMC24-Pass@32 | AIME25 | AIME25-Pass@32 | Avg |
|---|---|---|---|---|---|---|---|---|---|---|---|---|
| Qwen3-1.7B | - | 8K | 16.1 | 39.1 | 52 | 76.5 | 58.4 | 36.7 | 54.8 | 18.6 | 26.7 | 42.1 |
| Qwen3-1.7B | GRPO | 8K | 24.9 | 55.3 | 62.7 | 86.9 | 63.3 | 46.9 | 59.6 | 21.4 | 39.9 | 51.2 |
| Qwen3-1.7B | KDRL | 8K | 26.1 | 68.9 | 64.5 | 89.2 | **63.5** | 47.2 | 65.6 | 23.3 | 47.9 | 55.1 |
| Qwen3-1.7B | RLAD (ours) | 8K | **31.3** | **71** | **69.1** | **92.3** | **63.5** | **49.3** | **71.5** | **24.2** | **48.2** | **57.8** |
| Qwen2.5-1.5B-DS | - | 8K | 18.2 | 51.6 | 57.4 | 92.5 | 58.5 | 36 | 60.7 | 17.6 | 34.1 | 44.7 |
| Qwen2.5-1.5B-DS | GRPO | 8K | 29.5 | 63.4 | 76.3 | 94.9 | **64.8** | 47.9 | 70.5 | 22.8 | 38.6 | 56.5 |
| Qwen2.5-1.5B-DS | KDRL | 8K | 30.8 | 67.1 | 77.3 | 94.5 | 64.0 | 47.5 | 70.7 | 24.2 | 45.6 | 57.9 |
| Qwen2.5-1.5B-DS | RLAD (ours) | 8K | **31.9** | **70.1** | **78.2** | **95** | 64.4 | **49.2** | **70.8** | **25.1** | **47.1** | **59.1** |

*Table 3.* Long-context math reasoning results for Post-trained models at 8K context lengths. Results are reported in terms of Pass@1 and Pass@32.

ent initialization regimes, we consider both *base* and *post-trained* model settings.

- **Base models (R1-Zero-like).** Motivated by R1-Zero ([Guo et al., 2025](#)), which shows that rule-based RL can elicit reasoning behavior directly from pre-trained LMs, we use Qwen3-Base students to isolate the effect of our RL+distillation design. We study two student–teacher configurations: (i) Qwen3-1.7B-Base (student) with Qwen3-8B (teacher), and (ii) Qwen3-8B-Base (student) with Qwen3-32B (teacher). All base-model experiments use a 30K maximum response length, i.e., the longest supported by Qwen3-Base under a 2K prompt budget, to stress-test long-horizon reasoning.

- **Post-trained models.** We additionally evaluate post-trained students: (i) Qwen3-1.7B and (ii) Qwen2.5-1.5B DeepSeek-distilled ([Guo et al., 2025](#)), paired with their 8B/7B counterparts as teachers. Since Qwen3 post-trained models already incorporate substantial math-oriented post-training (CoT SFT, RL, and distillation) ([Yang et al., 2025](#)), we observe limited headroom under very long-context RL (e.g., 32K) in preliminary experiments. We therefore evaluate post-trained student models under an 8K response budget, framing the task as complex math reasoning under constrained compute where algorithmic gains are easier to measure.

**Datasets and Setup.** We evaluate on standard mathematical reasoning benchmarks, including MATH500 ([Hendrycks et al., 2021b](#)), AMC23 ([AI-MO, 2024b](#)), AMC24 ([rawsh,

2024](#)), AIME24 ([AI-MO, 2024a](#)), and AIME25 ([OpenCompass, 2025](#)). For all datasets, we report Mean@32, computed as the average score over 32 decoding runs; for the harder benchmarks (AMC23/24 and AIME24/25), we additionally report Pass@32. All evaluations are conducted using vLLM ([Kwon et al., 2023](#)) with a temperature of 0.6 and top-$p$ of 0.95, under 8K or 32K context windows consistent with the corresponding training regime.

### 5.2.2. MAIN RESULTS

**Training dynamics.** Figure [2](#) shows the validation trajectory on AIME24 ([AI-MO, 2024a](#)) for the Qwen3-8B-Base model (R1-Zero-like setting), reported as Mean@32 and Pass@32 over training. Consistent with the logical reasoning results, RLAD converges faster and reaches higher Mean@32. We also observe that explicit-KL-style distillation like KDRL can exhibit noticeable training instability, likely due to objective coupling between reward maximization and the auxiliary KL regularizer, which often requires careful hyperparameter balancing. In contrast, RLAD incorporates distillation into a unified trust-region update, resulting in more stable optimization.

**Results.** As shown in Table [2](#), RLAD improves Avg by +1.9 for Qwen3-1.7B-Base (38.4 vs. 36.5 GRPO) and by +5.5 for Qwen3-8B-Base (66.5 vs. 61.0 GRPO). Gains concentrate on harder benchmarks: Pass@32 metrics show particularly large improvements, e.g., AIME24-Pass@32 increases from 77.8 to 85.4 and AIME25-Pass@32 from 48.5 to 66.4 for Qwen3-8B-Base, while simpler tasks like MATH500 see smaller deltas (68.2 to 68.6). For post-trained

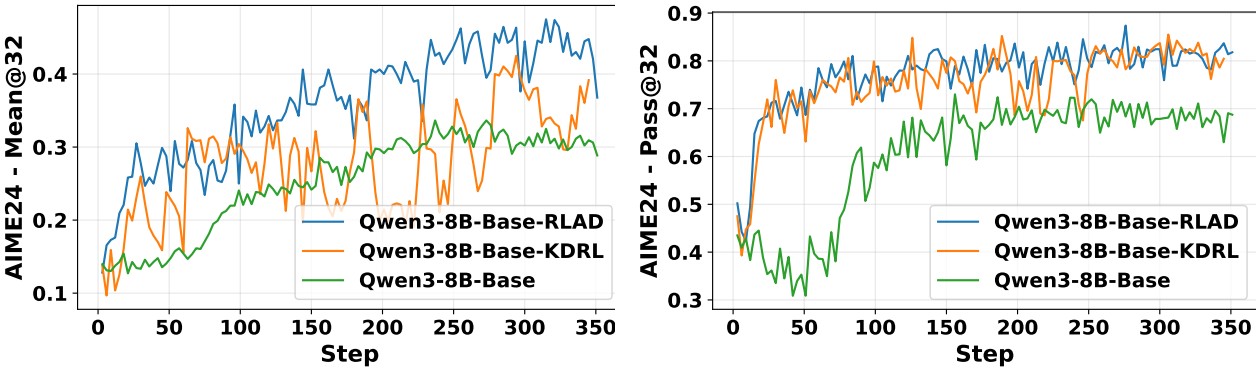

*Figure 2.* Comparison of validation accuracy on AIME-24 dataset dynamics during RL training on the math reasoning task for Qwen3-8B-Base, reported in terms of Mean@32 (as the mean Pass@1 over 32 runs) and Pass@32.

models (Table 3), the same pattern holds: RLAD raises Avg from 51.2 to 57.8 for Qwen3-1.7B and from 54.2 to 56.7 for Qwen2.5-1.5B-DS, with larger gains on AIME24/25 and more modest changes on MATH500. This suggests that algorithmic improvements are most impactful for difficult reasoning, while simpler tasks are closer to saturation and more constrained by model capacity, making the marginal gains from distillation comparatively smaller.

**Observation: RLAD Promotes Reward-Driven Policy Improvement over Pure Imitation.** Prior work (Yue et al., 2025a) observes that reinforcement learning typically yields larger gains in Pass@1 than in large-$K$ metrics, such as Pass@32. This behavior reflects the nature of RL optimization, which primarily improves the model's most probable behavior under its current policy, strengthening exploitation of existing competencies, rather than increasing the diversity of correct solutions emphasized by large-$K$ metrics. Consistent with this view, we observe a clear pattern across Figure 2 and Tables 2 and 3: distillation through KL regularization between the teacher and the student as in KDRL tends to produce larger relative gains in Pass@32 than in Pass@1 on challenging reasoning tasks. For instance, in Figure 2, KDRL achieves stronger improvements in Pass@32, while RLAD attains higher Mean@32 and more pronounced Pass@1 gains. This contrast suggests that the improvements of KDRL are driven primarily by stronger imitation of teacher solution modes, benefiting high-$K$ metric, rather than by reward-driven policy optimization. In contrast, RLAD more faithfully performs reinforcement learning: it prioritizes improving the student's reward-aligned behavior, yielding stronger Pass@1 gains and more stable training dynamics.

### 5.2.3. ABLATION STUDIES AND ROBUSTNESS

**Distillation coefficients.** Table 4 studies the sensitivity of KDRL to the KL loss weight $\lambda$ and RLAD to the mixture coefficient $\alpha$ on GSM8K and MATH. RLAD is stable across a broad range of $\alpha$, with noticeable degradation only near the extremes where one component dominates, while KDRL is more sensitive to $\lambda$. This supports using a fixed $\alpha = 0.5$ in the main experiments without extensive task-specific tuning.

**Weak-teacher robustness.** We further test whether teacher guidance can hurt when the teacher is weaker than the student. Using Qwen3-0.6B as the teacher for a Qwen3-1.7B student, Tables 5 and 6 show that RLAD stays close to GRPO, whereas KDRL degrades more noticeably. This aligns with the design of TRRD: the teacher shifts the trust-region center and clipping thresholds, but does not introduce an independent gradient term that can dominate the advantage direction.

**Clipping-ratio behavior.** To directly examine whether TRRD actively enforces a teacher-aware trust region, we measure the fraction of sampled tokens whose policy ratio hits the clipping boundary during Logic-RL training at 8K context. Table 7 shows that RLAD has substantially higher clipping frequency than GRPO, indicating that the teacher-centered trust-region constraint is active in the early update phase. The frequency decreases over training, suggesting that fewer sampled tokens violate the teacher-aware trust-region boundary as the student aligns.

**Robustness and variance.** As an additional robustness statistic, we report STD@32, the standard deviation over 32 decoding runs, for long-context math. Although this is not multi-seed training variance, it captures decoding-time robustness on the same long-context benchmarks. Table 8 shows that RLAD is comparable to or lower than GRPO on most settings and lower than KDRL on harder cases.

| Metric | GRPO | KDRL $\lambda = 10^{-3}$ | KDRL $\lambda = 10^{-2}$ | KDRL $\lambda = 10^{-1}$ | RLAD $\alpha = 0.1$ | RLAD $\alpha = 0.3$ | RLAD $\alpha = 0.5$ | RLAD $\alpha = 0.7$ | RLAD $\alpha = 0.9$ |
|---|---|---|---|---|---|---|---|---|---|
| GSM8K | 89.1 | 89.4 | 88.9 | 87.2 | 89.0 | **90.1** | 90.0 | 89.7 | 88.5 |
| MATH | 78.5 | 78.9 | 78.8 | 76.9 | 78.4 | 80.3 | **80.5** | 80.2 | 77.4 |

*Table 4.* Sensitivity of KDRL to the KL loss weight $\lambda$ and RLAD to the mixture coefficient $\alpha$ on GSM8K and MATH (Pass@1).

| Student | Teacher | Method | PPL2 | PPL3 | PPL4 | PPL5 | PPL6 | PPL7 | PPL8 | Avg |
|---|---|---|---|---|---|---|---|---|---|---|
| Qwen3-1.7B | - | GRPO | 1.00 | 0.99 | 1.00 | 0.98 | 0.95 | 0.92 | 0.81 | 0.950 |
| Qwen3-1.7B | Qwen3-0.6B | KDRL | 1.00 | 0.97 | 0.96 | 0.96 | 0.92 | 0.86 | 0.70 | 0.910 |
| Qwen3-1.7B | Qwen3-0.6B | RLAD (ours) | 1.00 | **1.00** | **1.00** | 0.97 | 0.94 | 0.91 | 0.80 | 0.946 |

*Table 5.* Weak-teacher ablation on K&K Logistics at 8K context. The student is Qwen3-1.7B and the weaker teacher is Qwen3-0.6B.

| Student | Teacher | Method | GSM8K | MATH |
|---|---|---|---|---|
| Qwen3-1.7B | - | GRPO | 89.1 | 78.5 |
| Qwen3-1.7B | Qwen3-0.6B | KDRL | 88.2 | 78.5 |
| Qwen3-1.7B | Qwen3-0.6B | RLAD (ours) | 89.1 | 78.3 |

*Table 6.* Weak-teacher ablation on GSM8K and MATH (Pass@1).

| Model | Training | Step 40 | Step 80 | Step 120 | Step 160 |
|---|---|---|---|---|---|
| Qwen3-0.6B | GRPO | 0.00194 | 0.00201 | 0.00176 | 0.00102 |
| Qwen3-0.6B | RLAD | 0.03022 | 0.02787 | 0.02252 | 0.01897 |
| Qwen3-1.7B | GRPO | 0.00285 | 0.00218 | 0.00196 | 0.00123 |
| Qwen3-1.7B | RLAD | 0.02881 | 0.02148 | 0.02252 | 0.01196 |

*Table 7.* Stepwise clipping frequency on Logic-RL at 8K context. Values report the fraction of sampled tokens that hit the policy-ratio clipping boundary.

| Model | Training | AIME24 | AMC23 | AMC24 | AIME25 |
|---|---|---|---|---|---|
| Qwen3-1.7B-Base | GRPO | 0.1454 | 0.1835 | 0.2306 | 0.1943 |
| Qwen3-1.7B-Base | KDRL | 0.2399 | 0.2046 | 0.2291 | 0.2047 |
| Qwen3-1.7B-Base | RLAD | 0.2082 | 0.1983 | 0.1563 | 0.1896 |
| Qwen3-8B-Base | GRPO | 0.1483 | 0.2086 | 0.1886 | 0.1490 |
| Qwen3-8B-Base | KDRL | 0.1562 | 0.2445 | 0.2814 | 0.1671 |
| Qwen3-8B-Base | RLAD | 0.1358 | 0.2213 | 0.2059 | 0.1398 |

*Table 8.* STD@32 for base models at 30K context. Lower values indicate smaller variation across 32 decoding runs.

| Model | Method | Updating Actor | Compute Log Probabilities | Total |
|---|---|---|---|---|
| Qwen3-8B-Base | GRPO | 28.9 | 24.5 | 491.2 |
| Qwen3-8B-Base | KDRL | 61.2 | 57.0 | 568.1 |
| Qwen3-8B-Base | RLAD | 60.9 | 56.2 | 567.4 |

*Table 9.* Latency comparison of RLAD and baselines in seconds when deploying teacher model and student model at the shared nodes. Total: total latency for updating a whole training batch (including multiple mini-batch and micro batches.)

### 5.2.4. EFFICIENCY ANALYSIS

We further analyze training efficiency when integrating RLAD into GRPO, compared with standard GRPO and KDRL, using Qwen3-8B-Base as the student and Qwen3-32B as the teacher. Table 9 reports the average wall-clock latency per training batch (global batch size 256, micro-batch size 16) with dynamic micro-batching enabled (Sheng et al., 2024), measured on 32 H200 GPUs. Although RLAD and KDRL deploy a 32B teacher, the teacher is only used to compute response of student rollouts for log-probabilities rather than generate additional trajectories. As a result, they incur only about 12% extra training latency, since the dominant cost in RL remains student rollout generation. Note that in these experiments the teacher is colocated on the same nodes as the student; if the teacher is served remotely, the overhead of RLAD and KDRL is expected to be even closer to standard GRPO. At inference time, RLAD requires no teacher model, so its evaluation efficiency is identical to the baselines.

## 6. Conclusion

We study reinforcement-aware knowledge distillation for reasoning and propose RLAD, a framework for RL post-training. RLAD replaces static KL-style teacher regularization with *Trust-Region Ratio Distillation* (TRRD), which integrates teacher guidance directly into advantage-weighted, trust-region policy updates. Across logical reasoning and long-context math reasoning tasks, RLAD consistently outperforms GRPO and KDRL, with the largest gains on challenging evaluations (e.g., AIME24/25 in Best@32) while remaining competitive on simpler tasks. By making distillation explicitly advantage-aware, RLAD reduces teacher–student mismatch and avoids interference between reward optimization and imitation. A current limitation is the reliance on teacher logits, which is most practical with open-weight, well-calibrated models. Extending TRRD-style distillation to closed-source teachers in RL settings remains an important direction for future work.

## Impact Statement

This paper presents work whose goal is to advance the field of Machine Learning. There are many potential societal consequences of our work, none which we feel must be specifically highlighted here.

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

| Metric | GRPO | KDRL | KDRL | KDRL | RLAD | RLAD | RLAD | RLAD | RLAD |
|---|---|---|---|---|---|---|---|---|---|
| | - | $\lambda = 10^{-3}$ | $\lambda = 10^{-2}$ | $\lambda = 10^{-1}$ | $\alpha = 0.1$ | $\alpha = 0.3$ | $\alpha = 0.5$ | $\alpha = 0.7$ | $\alpha = 0.9$ |
| GSM8K | 89.1 | 89.4 | 88.9 | 87.2 | 89.0 | **90.1** | 90.0 | 89.7 | 88.5 |
| MATH | 78.5 | 78.9 | 78.8 | 76.9 | 78.4 | 80.3 | **80.5** | 80.2 | 77.4 |

*Table 10.* Sensitivity of KDRL to the KL loss weight $\lambda$ and RLAD to the mixture coefficient $\alpha$ on GSM8K and MATH (Pass@1, test set).

## A. Ablation on Distillation Coefficients

We perform preliminary studies on two math reasoning benchmarks, GSM8K (Cobbe et al., 2021) and MATH (Hendrycks et al., 2021a), to examine the effect of the RLAD mixing coefficient $\alpha$ and the KDRL KL loss weight $\lambda$. Specifically, we train a Qwen3-1.7B student with GRPO, apply KDRL distillation using a Qwen3-8B teacher while varying $\lambda$, and apply RLAD distillation using the same teacher while varying $\alpha$. All models are trained on the training splits of GSM8K and MATH with a 1K context length, a learning rate of $1 \times 10^{-6}$, and a batch size of 1024. Table 10 reports Pass@1 accuracy for GRPO, KDRL, and RLAD under these hyperparameter settings. Overall, RLAD is largely insensitive to $\alpha$ across both benchmarks, with noticeable degradation only at extreme settings (e.g., $\alpha$ close to 0 or 1), while KDRL is more sensitive to the KL loss weight. We therefore set $\alpha = 0.5$ for all subsequent experiments.

## B. Stabilizing TRRD Training with Clipping

Unlike the standard PPO/GRPO setting—where the old and current policies are typically close—the teacher and student policies can be substantially mismatched early in training. According to prior works like TRPO (Schulman et al., 2015), this token-level ratio is a first-order Taylor approximation that holds only when teacher and student policies are close. According to our preliminary experiments, this mismatch can produce extreme distillation ratios in $\frac{\pi_{\theta^S}\left(y_t^{(i)}|x,y_{<t}^{(i)}\right)}{\pi_{\theta^T}\left(y_t^{(i)}|x,y_{<t}^{(i)}\right)}$, leading to outlier gradients and unstable optimization. To stabilize training, following PPO-style clipping (Schulman et al., 2017), we clamp the log-ratio $\log\frac{\pi_{\theta^S}\left(y_t^{(i)}|x,y_{<t}^{(i)}\right)}{\pi_{\theta^T}\left(y_t^{(i)}|x,y_{<t}^{(i)}\right)} \in [-1,1]$, which suppresses outliers and ensures smoother updates.

## C. TRRD as an Interpolation Between GRPO and DPO

The mixing coefficient $\alpha$ controls how much TRRD trusts the student's own old policy versus the teacher, while keeping the update driven by reward/advantage signals. When $\alpha = 1$, the anchor collapses to the student old policy and TRRD reduces to standard GRPO, recovering purely on-policy reinforcement learning. In contrast, as $\alpha \to 0$, the anchor approaches the teacher and the update becomes dominated by the teacher-referenced ratio $\pi_{\theta^S}/\pi_{\theta^T}$, yielding a KL-regularized imitation objective. This regime is closely related in spirit to DPO-style optimization (Rafailov et al., 2023), where policy updates are driven by log-probability ratios to a reference policy under an implicit reward/preference signal. Therefore, TRRD can be viewed as smoothly interpolating between on-policy GRPO and reference-policy-regularized preference optimization, with the teacher playing the role of the reference. In particular, as $\alpha \to 0$, TRRD induces a DPO-like objective by treating the teacher policy as the reference that defines implicit preferences.

Concretely, using the teacher as the reference, the DPO objective can be written as

$$\max_{\theta^S} \ \mathbb{E}_{x,\,y^+,\,y^- \sim \pi_{\theta^S}} \left[ \log \sigma\Big( \beta\big(\Delta \log \pi_{\theta^S}(x) - \Delta \log \pi_{\theta^T}(x)\big) \Big) \right],$$

where $\beta$ is the temperature, $\Delta \log \pi(x) = \log \pi(y^+|x) - \log \pi(y^-|x)$ and $(y^+, y^-)$ denotes a preference pair. The DPO logit is thus the *difference of log-probability gaps* between the student and the teacher on the same contrastive pair.

TRRD recovers this logit structure in the $\alpha \to 0$ limit. Taking logs of Eq. 3 yields, at the token level,

$$\log r^{\text{TRRD}} \ = \ \log \pi_{\theta^S}\left(y_t^{(i)} \mid x, y_{<t}^{(i)}\right) \ - \ \log \pi_{\theta^T}\left(y_t^{(i)} \mid x, y_{<t}^{(i)}\right).$$

Ignoring clipping, the surrogate in Eq. 4 becomes

$$\mathbb{E}\left[ \widehat{A}_{i,t}\big( \log \pi_{\theta^S} - \log \pi_{\theta^T} \big) \right],$$

which is a reward/advantage-weighted teacher-regularized objective. Aggregating token-level advantages into a pairwise preference signal leads to updates that depend on $\Delta \log \pi_{\theta^S} - \Delta \log \pi_{\theta^T}$, matching the DPO logit above. Despite this shared logit, the optimization dynamics differ: DPO is an *explicit pairwise* preference loss with sigmoid weighting, whereas TRRD remains an *on-policy* ratio/advantage-weighted update with GRPO-style clipping.

