# OpenReview forum: "Reinforcement-aware Knowledge Distillation for LLM Reasoning"
_ICML.cc/2026/Conference — ICML 2026 regular_

### Official Review · Reviewer_aBVS · 2026-02-20

**Soundness:** 3
**Presentation:** 2
**Significance:** 2
**Originality:** 3
**Overall Recommendation:** 4
**Confidence:** 4

**Summary:**

This paper proposes RL-Aware Distillation (RLAD), a framework that integrates teacher guidance into on-policy RL post-training for reasoning LLMs via Trust Region Ratio Distillation (TRRD). Across logical reasoning and long-context math with students from 0.6B to 8B and teachers up to 32B, RLAD shows consistent gains over GRPO, KDRL, and offline distillation.

**Compliance With Llm Reviewing Policy:**

Affirmed.

**Key Questions For Authors:**

Can you clarify whether TRRD is theoretically equivalent to a KL-regularized policy optimization objective with a dynamic reference policy?

**Limitations:**

see Weaknesses

**Strengths And Weaknesses:**

Strengths:
1. The method is simple to implement the standard GRPO, only needs teacher log-probabilities on student rollouts, no extra data collection.
2. The motivation is clearly articulated, and the selective imitation principle is compelling.
3. This paper offers an alternative to KL regularization that frequently interferes with reward optimization.

Weaknesses:
1. TRRD can be interpreted as KL regularization between the student, the previous policy, and the teacher. The method appears to a reformulation of trust-region policy optimization with an additional anchor rather than a fundamentally new distillation mechanism.
2. Improvements are measured only via task accuracy. It is unclear whether RLAD improves knowledge distillation or merely stabilizes reinforcement learning.

---

> ### Author Rebuttal · Authors · 2026-03-31
>
> Thank you for the helpful suggestions on strengthening the evidence for our mechanism. We clarify both points below.
>
> ---
>
> ## **Q1. Is TRRD equivalent to KL-regularized policy optimization with a dynamic reference?**
>
> We respectfully disagree with this characterization. TRRD admits a KL-style interpretation in the **unclipped** view, but it is not equivalent in optimization dynamics to KL-regularized RL.
>
> From Eq. (5),
> $$
> \log r^{\mathrm{TRRD}}=\alpha(\log \pi_\theta-\log \pi_{\mathrm{old}})+(1-\alpha)(\log \pi_\theta-\log \pi_T).
> $$
> Viewed in isolation, this resembles a weighted KL regularization toward the old policy and the teacher.
>
> However, the actual update is governed by the clipped PPO/GRPO surrogate
> $$
> \min\big(r^{\mathrm{TRRD}}A,\ \mathrm{clip}(r^{\mathrm{TRRD}},1-\epsilon,1+\epsilon)A\big).
> $$
> This is the key distinction. In TRRD, the teacher enters **inside** the clipped, advantage-weighted ratio, rather than as a separate teacher-KL term. Therefore, the teacher does not create an independent teacher-driven update direction; instead, it changes the anchor and clipping regime of a reward-driven update.
>
> More concretely, the advantage remains the gold standard: the sign of the sampled-token update is determined by the advantage, not by the teacher. The teacher can only modulate how large the update can be before saturation. Thus, when the teacher agrees with the reward signal, TRRD can amplify a useful update; when the teacher disagrees with the advantage, it does not redirect the update toward the teacher’s potentially wrong preference.
>
> By contrast, additive-KL KD introduces teacher guidance through a separate teacher-KL term whose effect is not conditioned on the sampled advantage. As a result, it can still pull the student toward the teacher even when that direction is not supported by the current reward signal.
>
> Therefore, while TRRD admits an unclipped KL-style interpretation, it is not equivalent in optimization behavior to adding a teacher-KL regularizer with a dynamic reference. A more accurate characterization is that TRRD is a **trust-region distillation mechanism**: reward determines the update direction, while teacher information reshapes the trust region around that reward-driven update.
>
> ---
> ## **Q2. RLAD improvement beyond task accuracy**
>
> Thank you for this helpful suggestion. We agree that analyses beyond task accuracy can further strengthen the paper. At the same time, we would respectfully note that task metrics such as Pass@1 and Pass@K are the standard primary measures for evaluating reasoning performance.
>
> That said, our evidence is not limited to accuracy alone:
>
> - **Robustness.** RLAD shows comparable or lower variance (STD@32) than GRPO, while KDRL exhibits higher variance, especially on harder long-context tasks (see **Reviewer KLpy Q3**).
> - **Mechanism evidence (clipping behavior).** RLAD exhibits substantially higher clipping frequency early in training, and this frequency decreases over time, which is consistent with a teacher-centered trust region being actively enforced at the token level and then violated less often as the student aligns (see **Reviewer 4Jpa Q2**).
> - **Weak-teacher behavior.** Under a weaker teacher, RLAD degrades gracefully and remains close to GRPO, whereas KDRL degrades more noticeably. This suggests improved robustness to imperfect teacher guidance rather than mere stabilization (see **Reviewer KLpy Q1–Q2**).
> - **Consistency and sensitivity.** RLAD consistently outperforms GRPO and KDRL across model sizes, tasks, and context lengths, while remaining stable across a broad range of $\alpha$ values (Appendix A; see also **Reviewer ytuK Q2**).
>
> Taken together, these results suggest that RLAD improves not only final task accuracy, but also the stability and quality of teacher-guided RL updates by making distillation advantage-conditioned. We will make this evidence more explicit in the revision and further expand the discussion beyond task accuracy in the final version.

---

> > ### Author Rebuttal · Reviewer_aBVS · 2026-03-31
> >
> > Thank you for your rebuttal. My concerns have been resolved and I will increase my score to 4.

---

> > > ### Author Response · Authors · 2026-04-07
> > >
> > > We thank the reviewer for the constructive feedback on clarifying the conceptual position of TRRD and strengthening the evidence beyond task accuracy. These suggestions have been valuable in sharpening the paper’s framing and analysis, and we will reflect them in the final version.

---

### Official Review · Reviewer_4Jpa · 2026-02-25

**Soundness:** 3
**Presentation:** 3
**Significance:** 3
**Originality:** 3
**Overall Recommendation:** 5
**Confidence:** 4

**Summary:**

This paper studies RL-aware KD for LLMs’ reasoning. The authors identify several limitations of the related RL+KD works and find that merging the reward and teacher’s supervision is the key to such combinations.

To address this issue, the paper proposes RL-Aware Distillation (RLAD), which integrates teacher guidance directly into the policy update through a modified importance ratio termed Trust Region Ratio Distillation (TRRD). Instead of adding a standalone KL term, TRRD anchors the PPO/GRPO trust region to a mixture of the teacher policy and the previous student policy, enabling advantage-aware and selectively activated imitation.

Experiments on logical and long-context mathematical reasoning benchmarks show that RLAD consistently outperforms standard GRPO and KL-based distillation (KDRL), with improved stability and larger gains on harder tasks. The paper is well-written and easy to follow. The proposed method is well-grounded and very effective. Although there are some limitations of the explanation and theory part, the current version is good enough for a positive score. I’m looking forward to seeing the improvement of this paper.

**Compliance With Llm Reviewing Policy:**

Affirmed.

**Key Questions For Authors:**

- For all the quantitative results, it would be helpful to also provide the performance of the teacher model. Then, it is easier to see the relative improvement and the potential gaps.
- Maybe a typo in equation-(4), it should be J^{RLAD}?

**Limitations:**

Yes

**Strengths And Weaknesses:**

## Strength:

- It is a very clever design to merge the teacher’s supervision with the π_ref to the ratio. Hence, the clipping can automatically incorporate the teacher’s judgment. The theoretical analysis also supports this claim quite well.
- A very clear explanation of why the proposed method is good compared with other baselines, especially the interplay between clipping ceiling and the teacher’s preference.
- Thorough experiments ranging from different models and datasets, the improvement is good.

## Weakness:

- The paper mentioned several times the shortcomings and limitations of other baselines. It would be more persuasive if the experiments could show some specific failure mode (e.g., serious target misalignment between the teacher’s supervision and reward signal, or big sensitivity of the hyper-parameter selection in KDRL, etc.) and how RLAD can mitigate them.
- The mechanism explanation of the interplay between clipping ratio and teacher’s preference is the highlight of the paper. Hence, I believe more quantitative results supporting this would be very helpful. For example, we can feed the same prompt and response to the original loss and RLAD’s loss, respectively. After that, we can count how many tokens’ gradients triggered the clipping mechanism. Then, a histogram (or scatter plot) showing how tokens with different π_S triggers clipping would be very cool.
- Regarding the training stability. By comparing different curves in Figures 1 and 2, it is difficult to conclude that RLAD is more stable than other baselines. Maybe having curves of several runs with different seeds would be helpful. (The step-wise trend of KDRL might be caused by a bad seed.)

---

> ### Author Rebuttal · Authors · 2026-03-31
>
> Thank you for the positive assessment and for the concrete suggestions on strengthening the evidence for our mechanism. We address each point below.
>
> ## **Q1. Specific failure mode when the teacher is weak or misaligned**
>
> Thank you for the suggestion. We agree that the paper should show a more explicit failure mode. Our added weak-teacher study does exactly this: with a weaker Qwen3-0.6B teacher for a Qwen3-1.7B student, RLAD stays close to GRPO, whereas KDRL degrades more noticeably. This is consistent with the mechanism of TRRD: the advantage remains the primary signal, and teacher guidance only modulates the update through the clipped trust-region ratio. Therefore, when teacher preference conflicts with the reward signal, RLAD is less affected by the teacher’s potentially wrong direction than additive-KL distillation. We will add this discussion more explicitly in the final version.
>
> ## **Q2. Quantitative clipping-ratio ablations**
>
> Thank you for the suggestion. As a quantitative proxy for the clipping analysis you proposed, we report the fraction of sampled tokens that hit the clip boundary during training. Since OpenReview does not support figures, we summarize stepwise statistics for Logic-RL at 8K in Table 7.
>
> **Table 7. Stepwise clipping frequency on Logic-RL (8K).**
>
> | Model | Training | Step 40 | Step 80 | Step 120 | Step 160 |
> |---|---|---:|---:|---:|---:|
> | Qwen3-0.6B | GRPO | 0.00194 | 0.00201 | 0.00176 | 0.00102 |
> | Qwen3-0.6B | RLAD | 0.03022 | 0.02787 | 0.02252 | 0.01897 |
> | Qwen3-1.7B | GRPO | 0.00285 | 0.00218 | 0.00196 | 0.00123 |
> | Qwen3-1.7B | RLAD | 0.02881 | 0.02148 | 0.02252 | 0.01196 |
>
> RLAD shows substantially more clipping events than GRPO, which is consistent with a teacher-centered trust region being actively enforced. The clipping frequency then decreases over training, suggesting that fewer sampled tokens violate the teacher-aware trust-region boundary as the student improves. We will add more detailed token-level analysis, including histograms/scatter plots, in the final version.
>
> ## **Q3. Robustness metrics and multi-run curves**
>
> Thank you for the suggestion. We add two clarifications.
>
> First, as an additional robustness statistic, we report STD@32 for long-context math (see Reviewer KLpy Q3). This is the standard deviation over 32 decoding runs, so it captures decoding robustness rather than multi-seed training variance. Under this metric, KDRL is generally higher-variance than RLAD on harder long-context tasks.
>
> Second, we agree that multi-run training curves would strengthen the stability claim. We have not completed full multi-seed long-context runs within the rebuttal timeline because of the compute cost, but we will add them in the final version. For now, we will tone down the wording in the paper to avoid over-claiming stability from single-run trajectories alone.
>
> ## **Q4. Teacher performance results**
>
> Thank you for the suggestion. We include teacher performance below and will add it to the final version.
>
> **Table 8. Teacher performance on the logistics task (8K).**
>
> | Model | PPL2 | PPL3 | PPL4 | PPL5 | PPL6 | PPL7 | PPL8 | Avg |
> |---|---:|---:|---:|---:|---:|---:|---:|---:|
> | Qwen3-1.7B | 1.00 | 0.99 | 1.00 | 0.98 | 0.95 | 0.92 | 0.81 | 0.950 |
> | Qwen3-8B | 1.00 | 1.00 | 1.00 | 0.99 | 0.98 | 0.94 | 0.87 | 0.969 |
>
> **Table 9. Teacher performance on complex math.**
>
> | Model | Context | AIME24 | AMC23 | MATH500 | AMC24 | AIME25 | Avg |
> |---|---|---:|---:|---:|---:|---:|---:|
> | Qwen3-8B | 8K | 65.6 | 88.1 | 89.4 | 64.7 | 56.0 | 72.8 |
> | Qwen3-8B | 30K | 75.7 | 93.8 | 97.2 | 71.6 | 67.1 | 81.1 |
> | Qwen3-32B | 8K | 72.1 | 92.1 | 94.3 | 67.1 | 57.3 | 76.6 |
> | Qwen3-32B | 30K | 81.4 | 94.7 | 97.5 | 74.9 | 72.9 | 84.3 |
>
> ## **Q5. Typos and misspellings**
>
> Thank you for catching this. We will correct Eq. (4) to $J^{RLAD}$ in the revision.

---

> > ### Author Rebuttal · Reviewer_4Jpa · 2026-03-31
> >
> > The authors' rebuttal has addressed most of my initial concerns. The added experimental results and discussions strengthen the paper. I hence increase my evaluation from 4 to 5. Looking forward to seeing the final version of the paper.

---

> > > ### Author Response · Authors · 2026-04-07
> > >
> > > We appreciate the reviewer’s positive feedback and the concrete suggestions on making the mechanism evidence more explicit. Your comments were particularly helpful in motivating additional quantitative analyses and presentation improvements, which we will incorporate in the final version.

---

### Official Review · Reviewer_ytuK · 2026-03-09

**Soundness:** 3
**Presentation:** 3
**Significance:** 2
**Originality:** 2
**Overall Recommendation:** 3
**Confidence:** 4

**Summary:**

This paper studies how to incorporate knowledge distillation into RL post-training for reasoning LLMs. It proposes RL-Aware Distillation (RLAD), which replaces an explicit teacher–student KL term with a trust-region ratio objective that integrates teacher guidance directly into policy optimization, and reports improvements over GRPO and KL-based distillation baselines on logic and math reasoning tasks.

**Compliance With Llm Reviewing Policy:**

Affirmed.

**Key Questions For Authors:**

1. The paper may overstate the practical advantage of RLAD over KDRL. The current presentation suggests that KDRL is categorically inferior due to its explicit KL regularization. However, based on my own training experience, KDRL can match RLAD under suitable configurations (e.g., on-policy training, batch size 64, rollout 8, and α=0.5). This raises the possibility that the reported gap is not intrinsic to the objective design, but instead depends on the specific training setup and hyperparameter choices.

2. The paper’s empirical comparison uses a modified KDRL baseline with log-ratio clamped to [-1,1] for training stability, which deviates from the standard KDRL implementation. More critically, the paper fails to demonstrate whether KDRL underwent equally thorough hyperparameter tuning as RLAD. This makes it impossible to determine if RLAD’s observed performance gains stem from a fundamentally superior objective function or merely from differences in training stability and hyperparameter tuning sensitivity between the two methods.

3. For post-trained models, the paper switches the training context from 32K to 8K simply due to limited performance headroom at 32K in preliminary experiments. This setting selection appears to be for the sole purpose of presenting favorable experimental results rather than aligning with practical application scenarios, lacking objective justification.

4. In the weak teacher scenario (where the teacher’s reasoning performance is inferior to the student’s), can RLAD’s selective imitation mechanism effectively avoid the misleading guidance from the weak teacher? Furthermore, is it feasible to reverse the distillation process and optimize the teacher using the student’s superior strategies (i.e., mutual distillation)?

5. What are the performance of KDRL and RLAD when the teacher and student models have the same parameter scale (e.g., 1.5B/7B student paired with a 1.5B/7B teacher)?

6. Have the authors tested cross-family teacher–student pairs, such as a Llama student with a Qwen teacher? Since different model families may exhibit different token distributions and reasoning styles, it would be valuable to know whether RLAD remains effective in this setting.

**Strengths And Weaknesses:**

The paper addresses an important problem, and the proposed idea is simple, intuitive, and well motivated. The method is reasonably novel, easy to implement on top of existing GRPO-style training, and the experiments cover multiple reasoning tasks and baselines with overall strong empirical results.

---

> ### Author Rebuttal · Authors · 2026-03-31
>
> Thank you for the thoughtful and constructive feedback. We clarify our claims below.
>
> ---
>
> ## **Q1. Practical advantage of RLAD over KDRL**
> We do not claim KDRL is categorically inferior; it is a strong baseline. Our claim is narrower: RLAD changes where teacher guidance enters the optimization. KDRL adds teacher guidance as a separate KL term, whereas TRRD incorporates it directly into the clipped trust-region ratio. As a result, teacher influence in RLAD is advantage-conditioned, while in additive-KL methods it is not. Empirically, this difference is most visible in weak-teacher settings and in sensitivity to hyperparameters.
>
> ---
>
> ## **Q2. Hyper-parameter comparison**
>
> To address fairness directly, we now ablate both KDRL’s KL weight $\lambda$ and RLAD’s interpolation coefficient $\alpha$ (Table 4). In our experiments, RLAD is less sensitive to $\alpha$ than KDRL is to $\lambda$.
>
> **Table 4. Sensitivity ablations (Appendix A).**
>
> | Metric | GRPO | KDRL ($10^{-3}$) | KDRL ($10^{-2}$) | KDRL ($10^{-1}$) | RLAD (0.1) | RLAD (0.3) | RLAD (0.5) | RLAD (0.7) | RLAD (0.9) |
> | --- | ---: | ---: | ---: | ---: | ---: | ---: | ---: | ---: | ---: |
> | GSM8K | 89.1 | 89.4 | 88.9 | 87.2 | 89.0 | **90.1** | 90.0 | 89.7 | 88.5 |
> | MATH | 78.5 | 78.9 | 78.8 | 76.9 | 78.4 | 80.3 | **80.5** | 80.2 | 77.4 |
>
> We will extend this comparison to longer-context tasks in the final version.
>
> ---
>
> ## **Q3. Ratio clipping in KDRL**
>
> Thank you for pointing this out. The sentence in the page-5 *Stability Considerations and Hyperparameter Ablations* paragraph is a drafting error in the manuscript and will be corrected in the revision. Its intended purpose was to emphasize the **ratio clipping used in TRRD / RLAD**, not to suggest clipping in KDRL.
>
> To clarify, KDRL does **not** define a PPO-style teacher–student importance ratio; its teacher guidance enters through a separate KL term. Therefore, there is no teacher–student ratio to clip in KDRL. Clipping is used only in **TRRD (RLAD)**, as described in **Appendix B**, where it is applied to the TRRD ratio within the PPO/GRPO-style surrogate.
>
> We will revise the manuscript to make this distinction explicit.
>
> ---
>
> ## **Q4. Why not evaluate at 32K**
>
> We now report 32K results (Table 5). For these post-trained Qwen3 models, GRPO itself yields only marginal gains at 32K, indicating limited headroom for further RL-based improvement. This is why we focus on more diagnostic settings such as base models at 30K and post-trained models at 8K.
>
> **Table 5. 32K results.**
>
> | Model | Training | Context | AIME24 | AMC23 | MATH500 | AMC24 | AIME25 | Avg |
> |---|---|---|---:|---:|---:|---:|---:|---:|
> | Qwen3-1.7B | - | 32K | 48.3 | 83.8 | 93.4 | 47.2 | 36.3 | 61.8 |
> | Qwen3-1.7B | GRPO | 32K | 48.5 | 83.7 | 92.9 | 47.4 | 36.8 | 61.9 |
> | Qwen3-8B | - | 32K | 75.6 | 93.8 | 97.4 | 71.7 | 67.0 | 81.1 |
> | Qwen3-8B | GRPO | 32K | 76.1 | 94.1 | 97.5 | 72.1 | 67.3 | 81.4 |
>
> ---
>
> ## **Q5. Weak teachers and mutual distillation**
>
> Please see **Reviewer KLpy Q2** for details. In summary, RLAD stays close to GRPO, whereas KDRL degrades more noticeably, suggesting that RLAD is more robust when teacher guidance is imperfect. Mutual distillation is technically feasible, but requires updating the teacher using teacher rollouts or alternating policies; we leave this to future work.
>
> ---
>
> ## **Q6. Same-scale teacher-student**
>
> RLAD is not restricted to large teacher-student gaps. In the same-scale setting (Qwen3-1.7B teacher for Qwen3-1.7B-Base student), both KDRL and RLAD improve over GRPO, with RLAD performing best (Table 6).
>
> **Table 6. Same-scale setting (30K).**
>
> | Model | Training | Teacher | AIME24 | AIME24-Pass@32 | AMC23 | AMC23-Pass@32 | MATH500 | AMC24 | AMC24-Pass@32 | AIME25 | AIME25-Pass@32 | Avg |
> |---|---|---|---:|---:|---:|---:|---:|---:|---:|---:|---:|---:|
> | Qwen3-1.7B-Base | - | - | 2.1 | 20.8 | 16.3 | 70.8 | 29.8 | 8.2 | 44.6 | 0.9 | 6.2 | 22.2 |
> | Qwen3-1.7B-Base | GRPO | - | 10.0 | 30.8 | 41.2 | 83.8 | 54.3 | 19.7 | 58.6 | 5.7 | 24.2 | 36.5 |
> | Qwen3-1.7B-Base | KDRL | Qwen3-1.7B | 11.2 | 33.6 | 41.5 | 82.7 | 58.7 | 21.9 | 61.5 | 7.8 | 24.9 | 38.2 |
> | Qwen3-1.7B-Base | RLAD | Qwen3-1.7B | 12.9 | 35.2 | 41.7 | 82.9 | 61.1 | 26.2 | 64.1 | 8.7 | 25.7 | 39.8 |
>
> ---
>
> ## **Q7. Cross-family teacher-student**
>
> We have not yet tested cross-family pairs such as Llama-to-Qwen. The main challenge is tokenizer/vocabulary mismatch: RLAD and KDRL require token-level log-probabilities on student rollouts, but tokens and vocabularies no longer align one-to-one, making KL/ratio objectives ill-defined. Approximate alignment (e.g., retokenization or sequence-level matching) introduces additional noise, which is especially problematic in RL. We note emerging work such as [1], and view this as an important direction for future work.
>
> **[1] Towards Cross-Tokenizer Distillation: the Universal Logit Distillation Loss for LLMs**

---

> > ### Author Rebuttal · Reviewer_ytuK · 2026-04-03
> >
> > Thank you for your reply. The experiments are primarily based on the Qwen3 model. I am curious about the performance of KDRL and TRRD under long chain-of-thought settings, for example, using DeepSeek-Distill-Qwen-1.5B as the student model with a teacher model such as JustRL-1.5B.

---

> > > ### Author Response · Authors · 2026-04-07
> > >
> > > Thank you very much for the follow-up. We would like to respectfully clarify that the paper is not limited to Qwen3. Beyond the Qwen3 results, we already evaluate a Qwen2.5-based long-CoT student, **Qwen2.5-1.5B-DS (DeepSeek-distilled)**, in the post-trained setting (Please see in Table.3 of the original paper). In the reported 8K setting, RLAD outperforms both GRPO and KDRL for this student, improving the Avg from **57.9** (KDRL) to **59.1** (RLAD). This suggests that the benefit of RLAD is not specific to Qwen3, but also transfers to a Qwen2.5/DeepSeek-distilled long-CoT student. For clarity, the current Table 3 in the paper contains a duplicated “GRPO” label; the second such row should correspond to **KDRL**, and we will correct this typo in the final version. We sincerely apologize for any confusion this may have caused.
> > >
> > > Beyond the 8K results already reported in the paper, we also ran new 16K experiments under the same training recipe (consistent to KDRL), and we report them in **Table 10** below. The trend remains consistent: RLAD continues to outperform KDRL, improving the Avg from **71.1** to **72.6**, with gains on AIME24, AMC23, AMC24, and AIME25. These additional results further support that RLAD remains effective in longer-CoT settings beyond the Qwen3 family.
> > >
> > > **Table 10. Qwen2.5-1.5B-DS long-CoT results (Skywork-OR1 recipe).**
> > >
> > > | Model | Training | Teacher | Context | AIME24 | AIME24-Pass@32 | AMC23 | AMC23-Pass@32 | MATH500 | AMC24 | AMC24-Pass@32 | AIME25 | AIME25-Pass@32 | Avg |
> > > |---|---|---|---|---:|---:|---:|---:|---:|---:|---:|---:|---:|---:|
> > > | Qwen2.5-1.5B-DS | KDRL | Qwen2.5-7B-DS | 8K  | 30.8 | 67.1 | 77.3 | 94.5 | 64.0 | 47.5 | 70.7 | 24.2 | 45.6 | 57.9 |
> > > | Qwen2.5-1.5B-DS | RLAD | Qwen2.5-7B-DS | 8K  | 31.9 | 70.1 | 78.2 | 95.0 | 64.4 | 49.2 | 70.8 | 25.1 | 47.1 | 59.1 |
> > > | Qwen2.5-1.5B-DS | KDRL | Qwen2.5-7B-DS | 16K | 42.1 | 81.7 | 82.7 | 96.0 | 90.3 | 62.1 | 85.2 | 29.7 | 69.8 | 71.1 |
> > > | Qwen2.5-1.5B-DS | RLAD | Qwen2.5-7B-DS | 16K | 43.9 | 84.5 | 84.6 | 96.2 | 90.2 | 63.7 | 86.1 | 31.8 | 72.4 | 72.6 |
> > >
> > > For the specific **DeepSeek-Distill-Qwen-1.5B + JustRL-1.5B** setting, we agree that this would be an informative additional comparison. However, JustRL is a later, stronger but also heavy recipe than the settings studied in our paper: the JustRL paper was released on **2025-12-18**, and the public training codes was released on **2026-02-05**, after the submission deadline. While the JustRL recipe appears stronger, it also requires longer training time (roughly **15 days on 32 GPUs**, according to the paper). This was not feasible within the discussion timeline. We will include this JustRL configuration, together with broader non-Qwen3 long-CoT comparisons, in our future version.

---

### Official Review · Reviewer_KLpy · 2026-03-13

**Soundness:** 3
**Presentation:** 3
**Significance:** 3
**Originality:** 3
**Overall Recommendation:** 5
**Confidence:** 4

**Summary:**

This paper studies how to distill reasoning ability from a stronger teacher model into a small student model during RL. The paper argues that the standard KD approach are not a good fit for this setting, offline distillation relies on a fixed trajectory from the teacher and does not match student's evolving on-policy rollouts, while an explicit teacher-student KL penalty will compete with reward maximization during training. To solve this, the paper proposes RL-aware distillation that, based on TRPO, modifies the PPO/GRPO-style loss so that the trust region is anchored to a mixture of the teacher's and student's old policies rather than the old student's policy itself. The proposed algorithm is designed to make imitation selective and advantage-aware, so that teacher guidance helps when it aligns with current policy improvement rather than acting as an unconditional auxiliary force.

**Compliance With Llm Reviewing Policy:**

Affirmed.

**Final Justification:**

The rebuttal addressed most of my concerns. The added clarification on the optimization mechanism, together with the additional ablations and robustness statistics, strengthens the paper. Some aspects could still be improved in the final version, but my overall assessment remains positive, and I will maintain my score of 5.

**Key Questions For Authors:**

My questions here are mainly about the ablation study and robustness.

1. Can the authors provide a more systematic ablation on the interpolation coefficient and teacher strength? Since TRRD is fundamentally defined by interpolating between the old student policy and the teacher policy, I would like to know how robust the gains are across different values of the mixture coefficient and across different teacher–student gaps. A strong answer here would increase my confidence that the proposed method is broadly useful rather than tuned to a narrow regime.

2. Can the authors clarify more sharply what TRRD achieves that a carefully tuned explicit-KL distillation baseline cannot? The paper argues that integrating teacher guidance into the trust-region update avoids objective interference, which is plausible. However, I would like a more explicit conceptual or empirical isolation of this effect. For example, does TRRD exhibit better token-level update alignment, better stability, or lower sensitivity to hyperparameters?

3. Can the authors report run-to-run variance or additional robustness statistics?

**Limitations:**

yes

**Strengths And Weaknesses:**

Strength: The paper addresses an important problem, and the main method is conceptually strong. TRRD is not presented as just another auxiliary loss; instead, it changes the policy update logistic by replacing the usual old-policy anchor with a mixture of teach-old-student-policy anchor. The discussion of TRPD as both a trust-region update and an implicit weighted KL view is also helpful for understanding the mechanism. The empirical results are promising and fairly broad with baseline including GRPO, offline distillation, and explict-KL style on-policy distillation. The reported gains are strong on harder tasks and also more stable than KDRL-like baselines.

Weakness: My main concern is the theoretical analysis is more intuitive rather than mathematically proof. The paper gives a convincing mechanism story, but the evidence of why TRRD should be fundamentally preferable to a carefully tuned KL-based method is still indirect. I would have liked a sharper isolation of what gains specifically from the trust-region mixture rather than from the teacher guidance more generally. The second weakness is the central contribution; the mixture design could be deeper ablated. Since the method depends critically on interpolating between the old student and the teacher with a tunable coefficient, a more systematic picture of how sensitive the method is to this coefficient and to teacher quality is strongly needed. The paper's empirical story is positive, but not clear how robust it is.

---

> ### Author Rebuttal · Authors · 2026-03-31
>
> We thank the reviewer for the positive assessment and for requesting sharper theory and stronger robustness evidence. We address the three questions below.
>
> ---
>
> ## **Q1. Theoretical clarification of RLAD vs. KL-based KD**
>
> The key point is: the advantage is the gold standard; the teacher cannot create an update opposite to the advantage, only modulate its size.
>
> For a sampled token $y_t$ with history $h_t$, TRRD uses
> $$
> r_t(\theta)=\left(\frac{\pi_\theta(y_t|h_t)}{\pi_{\mathrm{old}}(y_t|h_t)}\right)^\alpha
> \left(\frac{\pi_\theta(y_t|h_t)}{\pi_T(y_t|h_t)}\right)^{1-\alpha}.
> $$
> Equivalently,
> $$
> \log r_t=\alpha(\log \pi_\theta-\log \pi_{\mathrm{old}})+(1-\alpha)(\log \pi_\theta-\log \pi_T),
> $$
> which gives a KL-style interpretation only in the unclipped view.
>
> The actual optimization is the clipped surrogate
> $$
> \ell_t=\min(r_tA_t,\ \mathrm{clip}(r_t,1-\epsilon,1+\epsilon)A_t).
> $$
> Let $s=\log \pi_\theta(y_t|h_t)$. Then, whenever the subgradient is nonzero,
> $$
> \mathrm{sign}\left(\frac{\partial \ell_t}{\partial s}\right)=\mathrm{sign}(A_t).
> $$
>
> Thus, the sign of the sampled-token update is determined by the advantage, not by the teacher. The teacher only changes the anchor and clipping regime, i.e., how large the update can be before saturation. When the teacher agrees with the reward signal, RLAD can amplify a useful update; when the teacher disagrees with the advantage, RLAD is not redirected by the teacher’s potentially wrong decision. When $A_t\approx 0$, the TRRD contribution also vanishes.
>
> By contrast, additive-KL KD adds a separate teacher term to the RL objective. That term is not conditioned on $A_t$, so it can still pull the student toward the teacher even when that direction is not supported by the current reward signal. Therefore, TRRD is not equivalent in optimization dynamics to adding a teacher-KL auxiliary loss.
>
> ---
>
> ## **Q2. Ablation: coefficient and weak teacher**
>
> We add two ablations.
>
> First, $\alpha$ sensitivity: RLAD is stable across a broad range of $\alpha$, with degradation only near extreme values where one component dominates. We will move this study from Appendix A to the main text.
>
> Second, weak-teacher robustness: using Qwen3-0.6B as teacher for Qwen3-1.7B, RLAD stays close to GRPO, whereas KDRL degrades more noticeably. This is consistent with the mechanism above: teacher guidance in RLAD is filtered through the advantage rather than added as a separate unconditional term.
>
> **Table 1. Weak-teacher ablation on Logic-RL at 8K context.**
>
> | Student | Teacher | Method | PPL2 | PPL3 | PPL4 | PPL5 | PPL6 | PPL7 | PPL8 | Avg |
> |---|---|---|---:|---:|---:|---:|---:|---:|---:|---:|
> | Qwen3-1.7B | - | GRPO | 1.00 | 0.99 | 1.00 | 0.98 | 0.95 | 0.92 | 0.81 | 0.950 |
> | Qwen3-1.7B | Qwen3-0.6B | KDRL | 1.00 | 0.97 | 0.96 | 0.96 | 0.92 | 0.86 | 0.70 | 0.910 |
> | Qwen3-1.7B | Qwen3-0.6B | RLAD | 1.00 | 1.00 | 1.00 | 0.97 | 0.94 | 0.91 | 0.80 | 0.946 |
>
> **Table 2. Weak-teacher ablation on GSM8K / MATH.**
>
> | Student | Teacher | Method | GSM8K | MATH |
> |---|---|---|---:|---:|
> | Qwen3-1.7B | - | GRPO | 89.1 | 78.5 |
> | Qwen3-1.7B | Qwen3-0.6B | KDRL | 88.2 | 78.5 |
> | Qwen3-1.7B | Qwen3-0.6B | RLAD | 89.1 | 78.3 |
>
> ---
>
> ## **Q3. Robustness / variance**
>
> As an additional robustness statistic, we report STD@32, i.e., the standard deviation over 32 decoding runs, for long-context math. While this is not multi-seed training variance, it is still informative: RLAD is comparable to or lower than GRPO on most settings and lower than KDRL on harder cases.
>
> **Table 3. STD@32 for base models at 30K context.**
>
> | Model | Training | AIME24 | AMC23 | AMC24 | AIME25 |
> |---|---|---:|---:|---:|---:|
> | Qwen3-1.7B-Base | GRPO | 0.1454 | 0.1835 | 0.2306 | 0.1943 |
> | Qwen3-1.7B-Base | KDRL | 0.2399 | 0.2046 | 0.2291 | 0.2047 |
> | Qwen3-1.7B-Base | RLAD | 0.2082 | 0.1983 | 0.1563 | 0.1896 |
> | Qwen3-8B-Base | GRPO | 0.1483 | 0.2086 | 0.1886 | 0.1490 |
> | Qwen3-8B-Base | KDRL | 0.1562 | 0.2445 | 0.2814 | 0.1671 |
> | Qwen3-8B-Base | RLAD | 0.1358 | 0.2213 | 0.2059 | 0.1398 |
>
> We will add multi-seed training variance in the final version.

---

> > ### Author Rebuttal · Reviewer_KLpy · 2026-04-03
> >
> > The rebuttal addressed most of my concerns. The added clarification on the optimization mechanism, together with the additional ablations and robustness statistics, strengthens the paper. Some aspects could still be improved in the final version, but my overall assessment remains positive, and I will maintain my score of 5.

---

> > > ### Author Response · Authors · 2026-04-07
> > >
> > > We thank the reviewer for the positive assessment and for the thoughtful suggestions on sharpening the theoretical clarification and robustness analysis. These comments have helped us strengthen both the mechanism discussion and the empirical presentation, and we will incorporate the discussed revisions in the final version.

---

### Decision · Program_Chairs · 2026-04-30

**Decision:**

Accept (regular)

**Comment:**

Overall, the reviewers are positive toward the work. While Reviewer ytuK raised concerns regarding the choice of base models, the authors’ follow-up responses have addressed most of these issues. Accordingly, I recommend acceptance.